# Pollutant Gases to Algal Animal Feed: Impacts of Poultry House Exhaust Air on Amino Acid Profile of Algae

**DOI:** 10.3390/ani14050754

**Published:** 2024-02-28

**Authors:** Seyit Uguz, Arda Sozcu

**Affiliations:** 1Department of Biosystems Engineering, Faculty of Agriculture, Bursa Uludag University, Bursa 16059, Turkey; 2Department of Biosystems Engineering, Faculty of Engineering and Architecture, Yozgat Bozok University, Yozgat 66200, Turkey; 3Department of Animal Science, Faculty of Agriculture, Bursa Uludag University, Bursa 16059, Turkey; ardasozcu@uludag.edu.tr

**Keywords:** pollutant gas, microalgae, animal feed, sustainability, amino acids, heavy metals

## Abstract

**Simple Summary:**

Algae, rich in proteins, lipids, vitamins, and minerals, stand out as valuable feed ingredients in animal nutrition. Recognizing their nutritional benefits and environmental potential, this study investigates the impact of poultry exhaust air and culture conditions on the amino acid profiles of various microalgae species. The research aims to reduce pollutant gases from poultry operations by producing algal animal feed. Results show that *Synechococcaceae* cultivated with BBM and DI water achieved the highest protein content, while *Scenedesmus* sp. cultivated with DI water exhibited the highest carbohydrate content. *Synechococcaceae* grown with DI water had the highest essential and nonessential amino acids, except for glutamic acid and glycine. These findings contribute to evaluating microalgae as a sustainable protein and amino acid source, emphasizing environmental and economic considerations in animal nutrition.

**Abstract:**

Algae provide a rich source of proteins, lipids, vitamins, and minerals, making them valuable feed ingredients in animal nutrition. Beyond their nutritional benefits, algae have been recognized for their potential to mitigate the negative environmental impacts of poultry production. Poultry production is crucial for the global food supply but contributes to environmental concerns, particularly in terms of ammonia and carbon dioxide gas emissions. This study emphasizes the importance of reducing greenhouse gas and ammonia production in poultry operations by utilizing algae species suitable for animal consumption, highlighting the need for sustainable feed sources. This study investigated the effects of poultry exhaust air and culture conditions on the amino acid profiles of three microalgae species, namely, *Scenedesmus* sp. (AQUAMEB-60), *Ankistrodesmus* sp. (AQUAMEB-33), and *Synechococcaceae* (AQUAMEB 32). The experiments were conducted in a commercial broiler farm in Bursa, Turkey, focusing on reducing pollutant gas emissions and utilizing poultry exhaust air in algae cultivation. The highest protein content of 50.4% was observed in the biomass of *Synechococcaceae* with BBM and DI water. *Scenedesmus* sp. had the highest carbohydrate content of 33.4% cultivated with DI water. The algae biomass produced from *Synechococcaceae* growth with DI water was found to have the highest content of essential and nonessential amino acids, except for glutamic acid and glycine. The arsenic, cadmium, and mercury content showed variations within the following respective ranges: 1.076–3.500 mg/kg, 0.0127–0.1210 mg/kg, and 0.1330–0.0124 mg/kg. The overall operating costs for producing 1.0 g L^−1^ d^−1^ of dry algal biomass with the existing PBR system were $0.12–0.35 L^−1^ d^−1^, $0.10–0.26 L^−1^ d^−1^, and $0.11–0.24 L^−1^ d^−1^ for *Scenedesmus* sp., *Ankistrodesmus* sp., and *Synechococcaceae*, respectively. The operating cost of producing 1.0 g L^−1^ d^−1^ of protein was in the range of $0.25–0.88 L^−1^ d^−1^ for the three algae species. The results provide insights into the potential of algae as a sustainable feed ingredient in animal diets, emphasizing both environmental and economic considerations. The results demonstrated a considerable reduction in the production costs of dry biomass and protein when utilizing poultry house exhaust air, highlighting the economic viability and nutritional benefits of this cultivation method.

## 1. Introduction

Poultry production plays a crucial role in meeting the global demand for human nutrition, with high-quality proteins being an essential diet nutrient. The sustainable development of the scale-breeding industry has been identified as crucial for ensuring economic and environmental sustainability in poultry production [1]. The quest for sustainable poultry production has led to exploring alternative feed materials that are non-genetically modified and without any chemical residue. The utilization of algae in poultry feed has attracted significant attention due to its potential to enhance the sustainability and nutritional quality of poultry production [2,3].

Algae, including macroalgae and microalgae, offer a rich source of proteins, lipids, vitamins, and minerals, making them potentially valuable feed ingredients for farm animals [4,5]. Recently, the possibility of using algae in poultry rations has been explored as a safe and sustainable means of partially replacing conventional protein sources, with potential benefits for animal health and performance [6]. Research has indicated that algae, such as brown, green, and red algae, possess nutritional value that could positively impact poultry production performance and meat quality [6,7,8]. Additionally, the potential of algae to replace soybean expeller in organic broiler diets has been investigated, suggesting that algae may maintain a desirable amino acid profile within the feed, thus promoting sustainability in poultry farming [9]. Moreover, feed enzymes have been identified as a means to enhance the nutritional value of algae and improve their suitability as partial replacements for conventional and unsustainable feed ingredients, such as corn [5]. The nutritional value of specific algae, such as *Spirulina* and *Laminaria digitata*, has been studied in the context of poultry feed, with findings indicating their potential to influence broiler production performance, meat quality, and lipid profile [5,10].

In addition to their nutritional value, algae have been recognized for their potential to reduce the environmental impacts of animal production [11,12]. Poultry production is a significant contributor to the global food supply, providing a readily available and affordable source of animal protein [13]. However, the environmental impact of poultry production, particularly the emission of ammonia and carbon dioxide gases, has become a growing concern [14,15]. Poultry manure storage is a significant source of ammonia and greenhouse gas emissions, including methane, carbon dioxide, and nitrous oxide, all of which contribute to global warming and environmental degradation [16]. Furthermore, the concentration of air contaminants, including ammonia and carbon dioxide, has been analyzed in poultry housing facilities, with implications for animal health and productivity [17,18].

Algae have been recognized for their capacity to accumulate dissolved metals without toxicity, making them valuable in removing heavy metals from wastes [19]. Microalgae can remove heavy metals such as arsenic (As), cadmium (Cd), chromium (Cr), lead (Pb), and mercury (Hg), which are potentially harmful, from the environment through various techniques such as biosorption and bioaccumulation [20]. Thus, the heavy metal content of microalgal biomass could be mainly influenced by environmental conditions. Furthermore, the bioaccumulation properties of algae toward heavy metals necessitate careful evaluation of their potential impact, particularly in cases where algae are used for food, feed, or other applications [21].

Microalgae have the ability to synthesize many metabolites with rich nutrient content and biologically active substances, such as fatty acids and lipids, proteins and amino acids, carbohydrates, vitamins, minerals, pigments, etc. [2,22]. Therefore, it potentially provides a possibility for use as animal feed material or feed additive in animal nutrition. Previous studies have demonstrated that microalgae have high nutritional value for poultry, pigs, ruminants, and aquaculture [23,24]. The current findings highlight that microalgae biomass obtained from different species and grown with pollutant gases from poultry houses in different growing cultures have some advantages regarding different nutritional specifications as animal feed.

The usage of microalgae as a food or feed ingredient and their potential to accumulate heavy metals could negatively affect human or animal health if used without notification of the heavy metal content of algae biomass. Large-scale microalgae cultivation utilizes cost-saving efforts, for example, the usage of flue gas as a CO_2_ source or wastewater as a nutrient source. These waste sources are rich in heavy metals, which will accumulate in the microalgae. Studies have shown that different types of algae exhibit varying capacities for the bioaccumulation of heavy metals, with some species showing a higher preference for certain metals over others [25]. Napan et al. [26] investigated the impact of the most common heavy metals found in flue gas and their occurrence in powder of *Scenedesmus obliquus* biomass and highlighted that the presence of heavy metals at initial concentrations provided an improvement in growth and lipid production, but microalgal growth was limited at higher concentrations. Since most heavy metals were adsorbed by microalgal cells, the obtained biomass sample had a rich content of As, Cd, and Cr and, therefore, was not suitable for use as animal feed.

The utilization of poultry house exhaust air into algal cultivation systems could potentially introduce heavy metals found in exhaust air (the term “heavy metals” is commonly used to describe hazardous metals and metalloids in the literature). Heavy metals present in the exhaust air can be taken up by the algae and can accumulate in the biomass due to the permeable nature of algal cell walls, molecules, and ions in aqueous solutions to freely move through them. Additionally, heavy metals have the ability to attach to ligands found in these cell walls.

Today, commercial production of microalgae cannot be cost-competitive because it demands more water, energy, and nutrients [27,28]. Therefore, integrating microalgae cultivation systems and agricultural and industrial processes that release carbon dioxide and nutrients as waste could be potentially advantageous for creating a win–win situation that supports microalgae sustainability and environmental protection [29,30]. This study highlights the environmental impact of poultry production, focusing on reducing greenhouse gas and ammonia production by algae species suitable for animal nutrition, underscoring the need for sustainable feed sources in poultry farming. Furthermore, the economic viability of algae production has been underscored, particularly in utilizing algae biomass for animal feed, highlighting the potential for pollutant gases released from the poultry houses, for example, ammonia, nitrous oxide, carbon dioxide, hydrogen sulfide, and methane [31,32]. This study focused on the possible usage of the poultry exhaust air and different culture conditions (BBM, BBM−N, DI water) to culture microalgae and to compare the various parameters, including algae productivity, chemical composition, and cost analysis of three different algae species—namely, *Scenedesmus* sp. (AQUAMEB-60), *Ankistrodesmus* sp. (AQUAMEB-33), and *Synechococcaceae* (AQUAMEB 32)—and, subsequently, the possibility usage of this microalgal biomass as a feed ingredient in animal nutrition.

## 2. Materials and Methods

### 2.1. Algae Cultivation and Medium Conditions

*Ankistrodesmus* sp. (AQUAMEB-33), *Scenedesmus* sp. (AQUAMEB-60), and *Synechococcaceae* (AQUAMEB-32) were used because of their high growth rate with NH_3_ as the nitrogen source [33]. All three species were obtained from the AQUAMEB Culture Collection of Algae and Cyanobacteria (Aquatic Microbial Ecology and Biotechnology Laboratory, Bursa, Turkey). Each strain was processed for cultivation by transferring 1-L Erlenmeyer flasks containing 250 mL of Bold’s Basal medium (BBM) [33,34] that consisted of K_2_HPO_4_ (175 mg L^−1^), CaCl_2_·2H_2_O (25 mg L^−1^), MgSO_4_·7H_2_O (75 mg L^−1^), NaNO_3_ (250 mg L^−1^), K_2_HPO_4_ (75 mg L^−1^), MoO_3_ (1.42 mg L^−1^), NaCl (25 mg L^−1^), EDTA (50 mg L^−1^), KOH (31 mg L^−1^), FeSO_4_·7H_2_O (4.98 mg L^−1^), H_2_SO_4_ (0.1 µL L^−1^), H_3_BO_3_ (11.42 mg L^−1^), and a trace metal solution (1 mL L^−1^). The trace metal solution was composed of ZnSO_4_·7H_2_O (8.82 g L^−1^), MnCl_2_·4H_2_O (1.44 g L^−1^), CuSO_4_·5H_2_O (1.57 g L^−1^), CO(NO_3_)_2_·6H_2_O (0.49 g L^−1^). The medium pH was adjusted to 6.8–7.0 with 0.5 M HCl or 0.5 M NaOH, and before using this medium, it was subjected to sterilization in an autoclave for 20 min at 121 °C. The Erlenmeyer flasks were preserved in front of two daylight LED lights (ACK Lighting, İstanbul, Turkey). The prepared medium was transferred to flat plate PBRs with a 10-L working volume for testing in the poultry house.

### 2.2. Photobioreactors (PBRs) Set-Up

Algae cultures starting from 250 mL working volume were periodically doubled with autoclaved BBM and then transferred to 10 L PBRs as described by Uguz et al. [33] (Figure 1) upon reaching a volume of 10 L to test the effect of poultry exhaust air (mainly CO_2_ and NH_3_) and culture conditions (BBM, BBM−N, DI water). During the experiment, the PBRs were continuously vented with poultry exhaust air at the rate of 0.5 L min^−1^ per liter of PBR volume. The exhaust air was added to the PBR through a sparger bar placed at the bottom of the PBR. The NH_3_ and CO_2_ gas concentrations in the inlet and exhaust air of the PBR system were measured by a gas analyzer (Nenvitech-RS485/ 4–20MA CO_2_-NH_3_, Penta, İstanbul). The data logger device recorded gas concentrations every 5 min for 24 h throughout each experiment. Air flow meters (Bass Instruments, İstanbul, Turkey) were placed between the vacuum pump and the PBR to maintain the gas volumetric flow rate. Each algae species was grown in triplicate at a 200 μE m^−2^ s^−1^ light intensity and 24 h light photoperiod. The light intensity was determined using a LI-COR quantum sensor (LI-COR, Lincoln, NE, USA). Environmental conditions, including pH, temperature, water level (maintained with deionized water), aeration rate, and light intensity, were monitored throughout the experiments. The pH and temperature of the algal cultures were measured using a digital pH meter (HI98128, Hanna Instrument, İstanbul, Turkey). Each PBR was worked for 30 days under these conditions. During the experimental period, pH, temperature, cell count, and biomass were measured every three days.

### 2.3. Poultry House Experiment Design

Nine lab-scale PBRs were installed in a commercial, mechanically ventilated broiler farm located in Balıkesir, Turkey. The building measured 80 × 32 m and had suitable conditions with the commercial farms in this region. The capacity of the house was 42,000 birds for a 42-day growing cycle. In the study, nine experiments were conducted with three different algae species and three different medium conditions. Three experiments with three replicates run simultaneously for each species, including BBM, BBM−N, and DI water. The volume of each PBR was 10 L, containing 5 L inoculum and 5 L medium. The PBRs started with a BBM−N medium containing no nitrates. Poultry house exhaust air was the sole nitrogen source during the BBM−N experiments. For the DI water experiments, the PBRs contained 5 L inoculum and 5 L autoclaved DI water without any medium chemicals. The PBRs run for BBM experiments started with the original BBM containing 0.25 gL^−1^ NaNO_3_.

Since the concentrations of pollutant gases in the indoor air in broiler houses were lower in the first 10 days, the experiments were started on the 10th day of the 42-day growing cycle in the poultry house where the study was conducted. The shape and size of PBRs, airflow rates, and lighting schedule were all identically provided for each experiment. During the experiment, the PBRs were continuously aerated with poultry exhaust air. The air inlet for the PBRs was placed 0.3–0.4 m above the litter to feed a higher NH_3_ concentration to the PBRs. The nine PBRs were operated for 30 days and monitored throughout the experiment. The autoclaved deionized water was periodically added to maintain the culture volume of PBRs during the 30-day experiment. At the end of each experiment, the collected biomass from the PBRs was centrifuged, and the pellets were preserved at −80 °C for the lyophilization process.

Three experiments involving one algal species were conducted during each broiler production period in the poultry house experiments. At the company where the field study was conducted, a broiler production period lasted about 60 days, including 45 days for rearing and 15 days for cleaning and preparation for the new production period. The poultry house experiments began in January 2023 with experiments involving *Scenedesmus* sp. Since the subsequent production period started in April, experiments involving the *Synechococcaceae* species were also conducted in April. In June, during the following production period, experiments involving *Ankistrodesmus* sp. were conducted. Consequently, the indoor environmental conditions of the poultry house and NH_3_-CO_2_ gas concentrations differed in the experiments for each species. Daily pH and temperature monitoring were performed during the poultry house experiments. In the experiments with *Scenedesmus* sp. species, the pH values of the culture medium varied between 7.7 and 7.92, while in the experiments involving *Ankistrodesmus* sp. and *Synechococcaceae* species, the pH values were higher, ranging between 8.2 and 9.2.

### 2.4. Analytical Methods

Algal samples harvested from the nine experiments were analyzed for dry algal biomass concentration (mgL^−1^), cell concentration (cells L^−1^), and dry biomass production (mgL^−1^ d^−1^). A detailed description was given by Uguz et al. [33]. In brief, the cell concentration was determined using hemocytometers under an optical microscope. The optical density (absorbance) of algal cultures at 750 nm (OD_750_) was measured using a Hach DR3900 spectrophotometer (Hach, Loveland, CO, USA). The dry algal biomass concentration was determined by vacuum-filtering a 25 mL algal sample and weighing the filter after a drying process in a vacuum drying oven at 80 °C for 3 h. The weight of the biomass was determined using Equation (1):(1)$$Dry\;biomass\;conc.\left(g/L\right)=\frac{Final\;weight\;\left(g\right)-Initial\;weight\;(g)}{sample\;volume\;\left(mL\right)\times 1\;(L)}\times 1000$$

The biomass productivity (*P*, mgL^−1^d^−1^) was calculated using Equation (2) [35]:(2)Pbiomass=∆x∆t
where Δ*x* is the dry biomass concentration (gL^−1^) within a cultivation time of Δ*t* (day). The productivity of proteins, carbohydrates, and lipids was calculated using the following Formula (3) [35]:(3)$$P(mgL^{-1}d^{-1})=P_{biomass}\times C_{f}$$
where *P* is the productivity of proteins, carbohydrates, and lipids (mgL^−1^d^−1^); *P*_biomass_ is the biomass productivity (mgL^−1^ d^−1^); *C_f_* is the final content of protein, carbohydrates, and lipids given as a percentage of dry weight.

The amount of energy used for the vacuum pump and lighting was determined by measuring the electric power. The calculation of electric power usage was determined as [35]:(4)$$E_{\left(kWh/day\right)}=P_{\left(W\right)}\times t_{\left(h/day\right)}/1000_{\left(W/kW\right)}$$
where *E* is the daily power consumption in kilowatt-hours (*kWh*), *P*(*w*) denotes the rated power of the equipment in watts (*W*), and *t* represents the length of usage in hours per day. The power consumption was converted into dollar amounts by applying the conversion rate of 12.7 cents per kilowatt-hour in Balıkesir, Turkey, in order to determine the cost in dollars.

A total of 27 samples of dry algal biomass were analyzed for the chemical content according to the procedures of the Association of Official Analytical Chemists [36]. Lipid content was determined with acid hydrolysis analysis [26]. The crude protein was analyzed by the Kjeldahl method, using 6.25 as a conversion factor to calculate the protein content of samples [27]. Carbohydrate composition was determined from their alditol acetates via gas-liquid chromatography (GLC) and by gas-liquid chromatography-mass spectrometry (GLC–MS) according to the procedure given by Oxley et al. [37].

### 2.5. Determination of Fatty Acids and Amino Acids

Fatty acid methyl esters (FAME) were measured using gas chromatography analysis of the lyophilized microalgae biomass samples. A flame ionization detector (Agilent Technologies Inc., Wilmington, NC, USA)-equipped gas chromatograph (Agilent 6890N Series, Raleigh, NC, USA) was utilized to evaluate the FAME. The temperature of the column was applied at 120 °C for 1 min and then heated to 175 °C with an increment of 10 °C/min and kept there for 10 min. at a temperature of 175 °C. It was continually heated from 175 to 210 °C with an increment of 3 °C/min, then maintained at 210° for 5 min, further heated from 210 to 240 °C with an increment of 5°C/min, and finally kept at 240 °C for 5 min. Helium was employed as the carrier gas flowing at a rate of 2 mL/min through an HP 88-MS capillary column (100 m × 0.25 mm i.d., 0.20 µm film thickness; Agilent Technologies Inc.). The helium gas and one microliter of FAME were injected together. The injector and flame ionization detector were maintained at temperatures of 250 and 280 °C, respectively. To determine FAME in the samples, the retention times of samples were compared with FAME standards (Mixture ME-100; Greyhound Chromatography and Allied Chemicals, Birkenhead, Merseyside, UK).

After completing cultivation on the 7th day, the samples of microalgae biomass were collected for the amino acid profile. The pellets were freeze-dried and kept at −40 °C in a lyophilizer until future analysis. To determine the amino acid content of samples, an Agilent LC-MS mass spectrometer (Agilent Technologies, Santa Clara, CA, USA) was used, and amino acid concentrations were determined using electrospray ionization (ESI) and multiple reactions monitoring (MRM). The analysis process was applied according to the given method by Uguz and Sozcu [35].

### 2.6. Heavy Metal Analysis

The dried microalgae samples were subjected to a grinding process; then, homogenized samples with an amount of 0.5 g were digested in 10 mL of nitric acid (HNO_3_) at a temperature of 160 °C. The solutions were then diluted with deionized water at 50 mL according to the analysis process described by [38]. The concentrations of the heavy metals—cadmium (Cd), arsenic (As), and mercury (Hg)—were analyzed using an inductively coupled plasma mass spectrometer (Agilent 7700 ICP-MS, Agilent Technologies, CA, USA) with ASX-500 Autosampler. The optimized instrument conditions and measurement parameters are listed in Table 1.

### 2.7. Determination of Color Characteristics

The color characteristics, including lightness (*L**), redness (*a**), yellowness (*b**), chroma (*C*), and hue value (*α*) of dried algal biomass (*n* = 5 samples per each experimental group) were determined using a spectrophotometer (Konica-Minolta, Osaka, Japan).

### 2.8. Statistical Analyses

A 3 × 3 factorial arranged in a completely randomized design was employed in this study, where algae species and culture conditions were tested together. The experiments were replicated over time, with the trial being the blocking factor. The nine experiments with three replicates for each test condition consisted of three levels of algae species × three levels of culture conditions following a completely randomized (control and test tanks) design. The results were expressed in the form of mean ± standard deviation. The least significant difference (LSD) Student’s comparison test was used to compare the differences between treatments where significant differences were observed. A two-factor analysis was conducted to examine the effects of individual factors (algae species and culture conditions) and their interactions. ‘Student’s *t*-test was performed to compare the growth parameters (dry biomass, biomass productivity, etc.) under different test conditions. Software package JMP 11 was used for all the statistical analyses. Analysis of percentage data was conducted after an arcsine transformation of the data. Differences were considered significant at *p* ≤ 0.05.

## 3. Results and Discussion

### 3.1. Comparison of Nutritional Composition of Microalgal Biomass

Significant differences were observed for biomass production and productivity values of *Scenedesmus* sp., *Ankistrodesmus* sp., and *Synechococcaceae* grown with poultry house exhaust air, as shown in Table 2. The biomass productivity was the highest in *Scenedesmus* sp., *Ankistrodesmus* sp., and *Synechococcaceae* grown with BBM−N culture (43.0, 41.8, and 46.7 mgL^−1^d^−1^, respectively, *p* < 0.0015). It has been indicated that exhaust gases have sufficient nutrients, including carbon, nitrogen, phosphorus, etc., to enhance microalgal growth [29]. In previous studies performed by Boonma et al. [39] and Packer [40], it was reported that *Scenedesmus* spp. and *Acutodesmus* spp. cultivated with exhaust gases had higher biomass productivity. The increment in productivity was attributed to the stimulating effect of carbon dioxide for microalgal growth, and a higher CO_2_ concentration in the exhaust air (1740 ppm) was reported, which was significantly higher than ambient CO_2_ concentration (400 ppm, [39]. According to the current findings, it has been obvious that the three species of microalgae grown with BBM−N culture had higher productivity, which means that this medium could potentially work for algal growth under cultivation with poultry exhaust gases. Figure 2 shows the algal growth curves from the poultry house experiments.

The carbohydrate productivity was found to be the highest with a value of 12.5 mgL^−1^d^−1^ in *Scenedesmus* sp. grown with BBM−N culture, whereas the lowest value (3.1 mgL^−1^d^−1^) was observed in *Synechococcaceae* grown with DI water culture (*p* < 0.0008). Generally, the carbohydrate content varies between 15 and 75% of dry biomass in different microalgae species according to the growing conditions [41]. The differences between the current result and previous reports could be attributed to the microalgae species, culture conditions, and the use of exhaust gas from poultry houses [42,43]. Additionally, there are differences between the species for synthesizing the polysaccharides and synthesizing amounts [44]. It is known that stimulating carbohydrate production could result in a limitation of biomass production [45]. However, in the current study, an increasing trend in carbohydrate productivity has been clearly observed with increased biomass productivity in *Scenedesmus* sp. grown with BBM−N culture.

These findings clearly indicate that each microalga responds in different physiological pathways against environmental conditions during the cultivation process. The carbohydrate productivity could be influenced by extrinsic and intrinsic factors during the cultivation period [46]. Previous reports have clearly indicated that CO_2_ concentrations, temperature, pH, light intensity, salinity, and nitrogen source significantly affect the productivity capacity of microalgae [47,48]. It has been highlighted that both pH and CO_2_ levels could be the most crucial factors [39,40] due to the higher formation ratio of carbonic acid in the medium when the CO_2_ level increases [49].

In the current study, BBM−N provided maximum biomass productivity in three microalgae species, independently of varying growth conditions. This could be attributed to the presence of nitrogen in BBM. These hypotheses are supported by previous findings reported by Zarrinmehr et al. [50] and Feng et al. [51]. On the other hand, *Scenedesmus* sp. had the highest productivity capacity for carbohydrates in BBM−N compared with the other microalgae species and BBM conditions.

Another critical issue is microalgae’s protein productivity and quality as an alternative to traditional protein sources used in animal nutrition. Van Krimpen et al. [52] indicated that the protein yield of microalgae is higher than that of conventional protein sources in North Western Europe according to the 2–25 factors. Modifying the medium composition is a direct method to enhance protein content in algae. For instance, increased CO_2_, phosphate, and nitrogen appear to promote algal protein accumulation [53]. Therefore, this study also investigated the protein productivity of microalgae grown using poultry house exhaust air. The PBRs in this study were fed with poultry exhaust air, which involves a high concentration of nitrogen-based gas ammonia that directly affects the algal protein productivity. The protein productivity was significantly higher in *Scenedesmus* sp. and *Synechococcaceae* grown with BBM−N culture (20.6 and 18.8 mgL^−1^d^−1^, respectively). The lowest value of protein conductivity ranged between 7.2 and 9.4 mgL^−1^d^−1^ in biomass obtained from *Scenedesmus* sp. grown with DI water and *Ankistrodesmus* sp. grown with all cultures (*p* < 0.0001). These findings clearly suggest that the capability of *Synechococcaceae* for protein production is much higher than that of the other two species of microalgae. This could potentially be important as an alternative protein source in both ruminant and poultry nutrition [54,55].

The nutrient composition of algae species cultivated with poultry house exhaust air under different culture conditions is shown in Table 3. The content of fatty acids is noticeably different among the microalgae species and culture conditions. The highest and lowest content of SAFA was observed in biomass obtained from *Ankistrodesmus* sp. grown with DI water (60.1%) and *Scenedesmus* sp. grown with BBM−N culture (30.8%), and *Scenedesmus* sp. grown with DI water (30.2%, *p* < 0.0001). The MUFA content was found to be the highest in *Scenedesmus* sp. grown with DI water (32.8%) and the lowest in *Synechococcaceae* grown with DI water (5.5%, *p* < 0.0001). The biomass obtained from *Scenedesmus* sp. grown with BBM−N culture had the highest content of PUFA (36.6%), whereas the lowest content of PUFA was observed in *Synechococcaceae* grown with BBM culture (5.2%, *p* < 0.0001).

Microalgae produce fatty acids, including saturated (SAFA), mono (MUFA), and polyunsaturated (PUFA) fatty acids [55]. Due to the higher energy requirement of SAFA absorption, a diet with a rich content of MUFA is desired in animal nutrition [56]. On the other hand, long-chain PUFA significantly benefit health by providing a preventive effect against diseases [57]. In this respect, it could be suggested that *Scenedesmus* spp. grown with BBM−N and DI water cultures could have the potential to be an alternative to vegetable or animal-based oil sources in animal nutrition due to their lower content of SAFA and higher content of MUFA and PUFA.

The protein and carbohydrate contents of algae biomass were affected by both algae species and culture conditions (*p* < 0.0001). *Synechococcaceae* grown with BBM and DI water had the highest protein content with a value of 50.4%, whereas *Ankistrodesmus* sp. grown with BBM−N and DI water cultures had the lowest content (20.2%). The carbohydrate content was found to be the highest in *Scenedesmus* sp. grown with DI water (33.4%) and the lowest in *Synechococcaceae* grown with DI water (14.0%, *p* < 0.0001). The protein content of microalgal biomass could vary within 40–60% of dry matter, according to some factors, including the species, cultivation methods, growing stage of microalgae, and harvesting time [58,59].

In animal nutrition, animal-originated protein sources have high-quality protein due to the well-balanced content of essential amino acids [60,61]. Nosworthy et al. [62] indicated that *Chlorella* spp. had a higher quality of protein when compared with the other pulses, such as beans, lentils, and peas. Related to protein content, current findings highlight that *Synechococcaceae* grown with BBM and DI water could be accepted as a high protein source content, potentially comparable to the protein level of conventional protein sources such as soybeans in poultry nutrition.

Microalgal biomass has rich carbohydrate content, also known as polysaccharides, and many types of structures of carbohydrates according to the many factors [63]. Moreira et al. [64] reported that the polysaccharides obtained from microalgae are safer, more stable, biocompatible, and biodegradable when compared with other polysaccharide sources. Today, different species of microalgae, for example, *Tetraselmis* sp., *Chlorella* sp., *Spirulina platensis*, *Scenedesmus* sp., *Chlorella vulgaris*, and *Haematococcus pluvialis*, are widely produced for carbohydrate extraction [64,65,66]. In the same way, current findings clearly demonstrate that *Scenedesmus* sp. grown with all types of cultures have a higher carbohydrate content, which could be an indicator for using it as a carbohydrate source, especially in ruminant nutrition. Because of the microbial degradation process in the rumen of ruminants, they have the ability to degrade the cell wall carbohydrates when compared with the monogastric animals, without feed additive supplementation of exogenous enzyme or some biodegradation process of feedstuffs, except for highly lignified feed materials [67].

### 3.2. Amino Acid Profile

The current results clearly show that the pathway of amino acid formation showed variations according to the microalgae species and culture conditions. The essential and nonessential amino acid composition (mg/100 mg of dried weight) is illustrated in Figure 3 and Figure 4. Arginine, histidine, isoleucine, leucine, methionine, phenylalanine, threonine, and valine contents were found to be the highest in biomass obtained from *Synechococcaceae* grown with DI water (2.88, 0.66, 1.50, 3.44, 1.10, 1.90, 2.62, and 1.98 mg/100 mg, respectively, *p* < 0.0001). On the other hand, the content of lysine was found to be higher in biomass obtained from *Synechococcaceae* grown with DI water and BBM−N cultures and, also, *Scenedesmus* sp. grown with BBM culture (1.58, 1.58, and 1.51 mg/100 mg, respectively, *p* < 0.0001). A higher content of alanine, asparagine, glycine, proline, serine, and tyrosine was observed in biomass obtained from *Synechococcaceae* grown with DI water, whereas the glutamine content was found to be the highest in *Scenedesmus* sp. grown with DI water, BBM−N, and BBM cultures (*p* < 0.0001).

Regarding the amino acid profile of microalgal biomass, it could be suggested that microalgae could have an opportunity to be an alternative to conventional protein sources in animal nutrition. This is similar to previous reports explained by Burja et al. [68], Jacob-Lopes et al. [69], and Uguz and Sozcu [35]. Regarding both essential and nonessential amino acids, *Synechococcaceae* grown with DI water could be offered as an amino acid source in animal nutrition. In a previous report, it was highlighted that different species of microalgae could be used in combination to provide a more balanced amino acid profile [62].

### 3.3. Heavy Metal Content

The heavy metal content of algae biomass from different microalgae species and cultivation practices is presented in Table 4. The results show that both microalgae species and culture conditions affected the ability of heavy metal accumulation of microalgae in a different manner. The content of arsenic and cadmium were found to be the highest, with a value of 3.500 mg/kg and 0.0121 mg/kg, respectively, in microalgal biomass obtained from *Ankistrodesmus* sp. grown with BBM−N culture (*p* < 0.001). The content of arsenic and cadmium showed variations between a range of 1.076–3.500 mg/kg and 0.0127–0.1210 mg/kg in the study (Table 4). This is in accordance with the regulations of EU Directive 2002/32/EC [70]. The maximum levels for arsenic, cadmium, and mercury in feed and feed materials according to the EU Directive 2002/32/EC (Regulating No 744/2012) are given as 40 mg/kg, 1 mg/kg, and 0.1 mg/kg, respectively. In the study, the content of mercury was found to be the highest in the microalgal biomass of *Synechococcaceae* grown with BBM−N culture (*p* < 0.001). This is a critical value and higher than the maximum limit given by EU Directive 2002/32/EC [70].

It is known that microalgae have the capability to absorb potentially harmful heavy metals through different pathways, including biosorption, biotransformation, or bioaccumulation [20]. Therefore, the heavy metal content of microalgae shows variations according to the cultivation conditions [71]. Due to the accumulation of heavy metals by microalgae, its usage in food or feed chains could cause adverse effects on human or animal health. In this respect, before applying microalgae biomass as a nutritional material, the analysis of heavy metals, mainly Pb, Cd, Ar, and Hg, must be provided and guaranteed for their safety [72]. However, Kay and Barton [73] hypothesized that heavy metals originating from microalgae could not be effectively absorbed by the gastrointestinal tract of animals. According to the current findings, *Scenedesmus* sp. and *Ankistrodemsus* sp. could be acceptable for heavy metal content, but *Synechococcaceae* grown with BBM−N culture has an exception for mercury level. Therefore, the analysis of microalgal biomass obtained from *Scenedesmus* sp. and *Ankistrodesmus* sp. in this study has indicated that the heavy metal content is well within the prescribed safety limits according to EU regulations for animal feed (Table 4).

### 3.4. The Color Characteristics of Algae

The color characteristics of algae biomass from different microalgae species and cultivation practices are presented in Table 5. The lightness (*L**) value was found to be the lowest, with a value of 30.8 in algae biomass produced from *Synechococcaceae* grown with BBM culture (*p* < 0.001). The highest mean of *a** value was observed in *Scenedesmus* sp. grown with BBM−N culture (−5.4), whereas *b** value was observed in *Ankistrodesmus* sp. grown with BBM−N culture (23.7, *p* < 0.0001).

In animal nutrition, the color characteristics, meaning the pigment content of feedstuffs, have importance due to their capability to change the color characteristics of animal products, for example, meat or egg yolk color. Especially the *a** value and *b** values indicate the red–green and yellow–blue color characteristics, respectively. Regarding current findings, *Scenedesmus* sp. grown with BBM−N had proximity for redness, whereas *Ankistrodesmus* sp. grown with BBM−N was close to yellowness. Interestingly, in our previous study [35], the color characteristics were rather different from current findings on account of darkness value (*L** value). It was lower than the value of 30.0 in the previous study of the same three microalgae species [35], and it was found to be higher than the mean value of 30.0 in this study. The variation of the color characteristics could be related to the harvesting time and culture conditions in the studies.

### 3.5. The Economic Estimates of the PBR System

Table 6 presents the cost estimates of the PBR system operated in the poultry house. Poultry house experiments were conducted using 10 L photobioreactors (PBRs) to investigate the impact of different culture strategies (BBM, BBM−N, DI water) and algae species on the amino acid and nutritional content of the algae. The capital cost of the system, including flowmeters, pumps, and PBR tanks, was $218.24 (Table 6). A 1.5 hp air pump was used to transfer the poultry house exhaust air to each PBR. Operating costs associated with PBR systems include energy consumption, nutrients, and operational management. All the energy consumed by the PBR system was for pumping and lighting. The cost of pumping (including aeration) was $0.034 L^−1^ d^−1^, while the lighting cost was $0.0024 L^−1^ d^−1^. These costs were calculated based on the power rate in Bursa, Turkey. The operating costs were $0.0824, $0.0524, and $0.0364 L^−1^ d^−1^ for BBM, BBM−N, and DI water culture conditions, respectively.

Dry biomass, protein, and carbohydrate production costs under three different culture conditions for *Scendesmus* sp., *Ankistrodesmus* sp., and *Synechococcaceae* are shown in Figure 5. The overall operating costs for producing 1.0 g L^−1^ d^−1^ of dry algal biomass with the existing PBR system were $0.12–0.35 L^−1^ d^−1^, $0.10–0.26 L^−1^ d^−1^, and $0.11–0.24 L^−1^ d^−1^ for *Scenedesmus* sp., *Ankistrodesmus* sp., and *Synechococcaceae*, respectively. *Ankistrodesmus* sp., grown with DI water had the lowest price for dry algal biomass, whereas *Scenedesmus* sp. grown with BBM had the highest price. The operational costs of batch and continuous cultivation modes for the same algae species in our previous study [35] conducted in laboratory conditions were $0.24–0.37 L^−1^ d^−1^, $0.34–0.6 L^−1^ d^−1^, and $0.47–0.95 L^−1^ d^−1^ for *Scenedesmus* sp., *Ankistrodesmus* sp., and *Synechococcaceae*, respectively. The operational costs of PBRs run in the poultry house were lower compared with our previous study.

The cost for producing 1.0 g L^−1^ d^−1^ of carbohydrate was $0.42–1.27 L^−1^ d^−1^, $0.39–1.58 L^−1^ d^−1^, and $0.7–1.37 L^−1^ d^−1^ for *Scenedesmus* sp., *Ankistrodesmus* sp., and *Synechococcaceae*, respectively. The cost for producing 1.0 g L^−1^ d^−1^ of protein was $0.25–0.74 L^−1^ d^−1^, $0.51–0.88 L^−1^ d^−1^, and $0.28–0.49 L^−1^ d^−1^ for *Scenedesmus* sp., *Ankistrodesmus* sp., and *Synechococcaceae*, respectively. The operating cost of producing 1.0 g L^−1^ d^−1^ of protein in our previous study was in the range of $1.6–3.7 L^−1^ d^−1^ for the same algae species. Producing microalgae with poultry house exhaust air resulted in a 58% and an 84% reduction in the dry biomass and protein production costs, respectively. The reason for the lower protein costs of microalgae produced in the poultry house experiments is due to the higher algal production efficiency in the PBR system fed with the poultry house indoor air. The protein contents of dry biomass produced with poultry house air were higher than those grown under laboratory conditions. Although the dry biomass productivity of microalgae grown with poultry house air is considerably lower than in laboratory conditions, the higher nutritional value makes the system more economical.

The costs are significantly higher compared with the other feed ingredients for poultry nutrition. It is essential to point out that the cost estimates were derived specifically for 10 L photobioreactors (PBRs). Acien et al. [74] reported that scaling up the PBR system by 2.2 times led to a significant (82%) decrease in algae production costs. Vazquez-Romero et al. [27] stated that the production scale increased from 294.84 to 2948.41 metric tons of harvested biomass per year when the PBR system expanded from 10 to 100 hectares. The production cost of biomass was reduced by 51.26% (52.77 EUR/kg DW of biomass) for a 10-hectare area and by 59.36% (44 EUR/kg DW of harvested biomass) for a 100-hectare area. In another study, Tredici et al. [75] similarly reported a comparable decrease from 1 to 100 hectares. Consequently, expanding the photobioreactor (PBR) system holds the potential for significant advancements.

## 4. Conclusions

The interest in microalgae as an alternative food and/or feed material is expected to increase due to the demand for plant-based products and the healthy nutritional characteristics of microalgae. It could be affirmed that the feed material obtained from microalgae could be efficiently used due to its nutritional composition through partial replacement of conventional dietary protein sources. Nevertheless, the nutrient availability of different microalgae species must be considered to determine their potential usage in animal nutrition. In that respect, *Scenedesmus* sp. could be suggested due to its lower content of SAFA and higher content of MUFA and PUFA, and, also, higher carbohydrate content, whereas *Synechococcaceae* could be offered for animal nutrition because of its high content of protein and both essential and nonessential amino acids profile. The heavy metal content of microalgal biomass was acceptable according to the EU Directive 2002/32/EC (Regulating No. 744/2012). It is an important safety issue in animal production. Therefore, *Scenedesmus* sp. has many more advantages for its nutritional content and safety usage and has the potential to be cultivated with exhaust gas.

Since the interest in microalgal biomass is growing in the sector, more research should be focused on different strategies for the sustainable and economic cultivation of microalgae in animal production. Novel technology should be explored to improve microalgal nutrient utilization, and their long-term nutritional and metabolic effects should be assessed.

## Figures and Tables

**Figure 1 animals-14-00754-f001:**
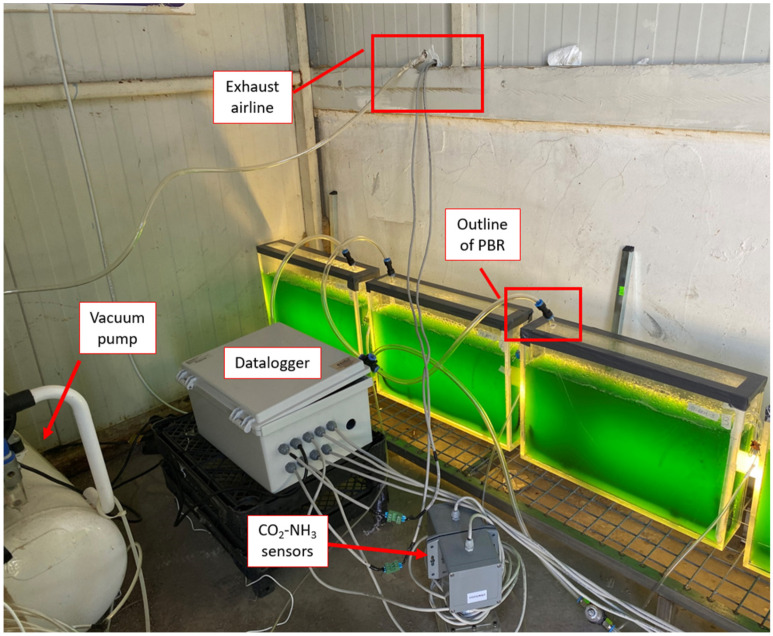
PBRs running in the poultry house.

**Figure 2 animals-14-00754-f002:**
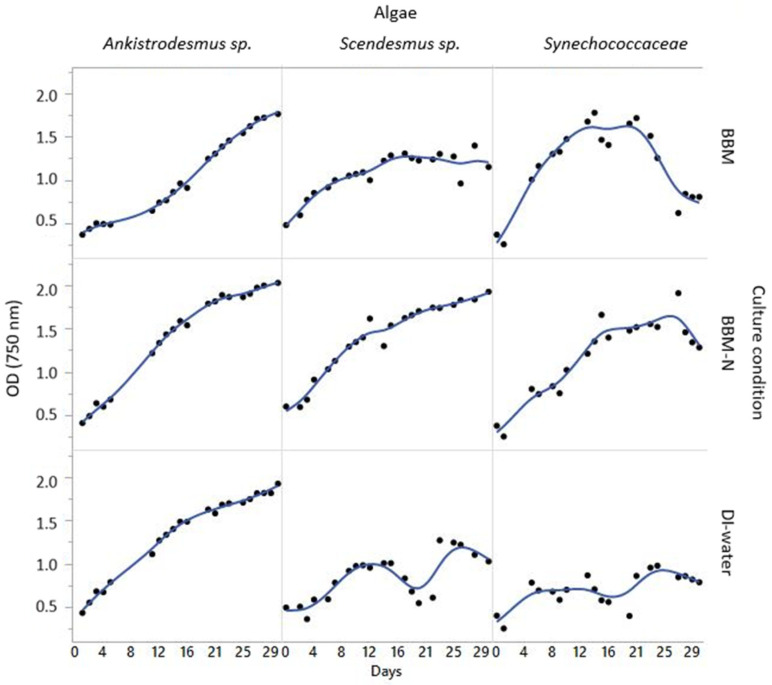
Algal growth curves from the poultry house experiments (OD = 750 nm).

**Figure 3 animals-14-00754-f003:**
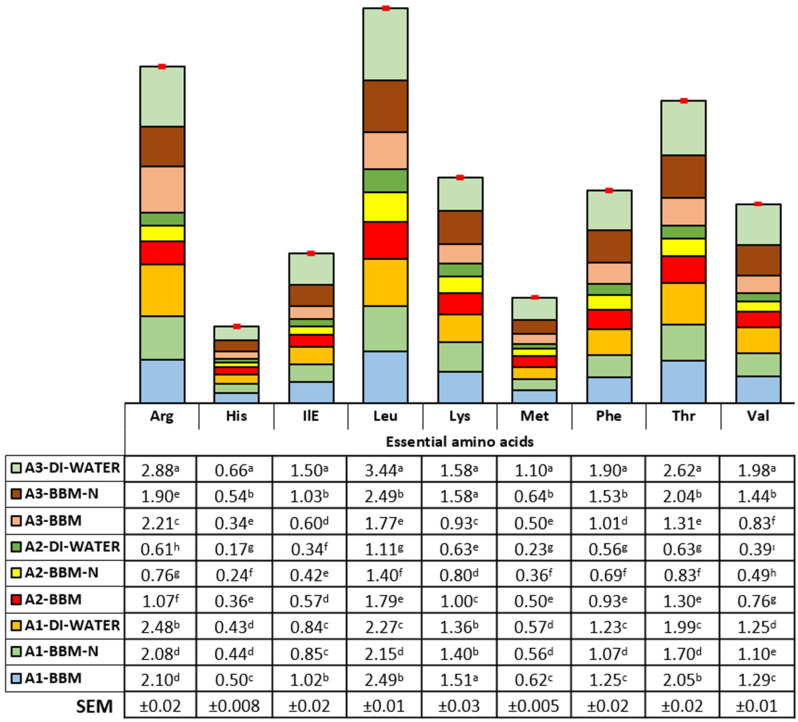
Essential amino acid composition (mg/100 mg) of algae species (*Scenedesmus* sp. (A1), *Ankistrodesmus* (A2), and *Synechococcaceae* (A3)) cultivated with poultry house exhaust air. ^a–i^ Differences in letters within columns indicate significant differences among the experimental groups.

**Figure 4 animals-14-00754-f004:**
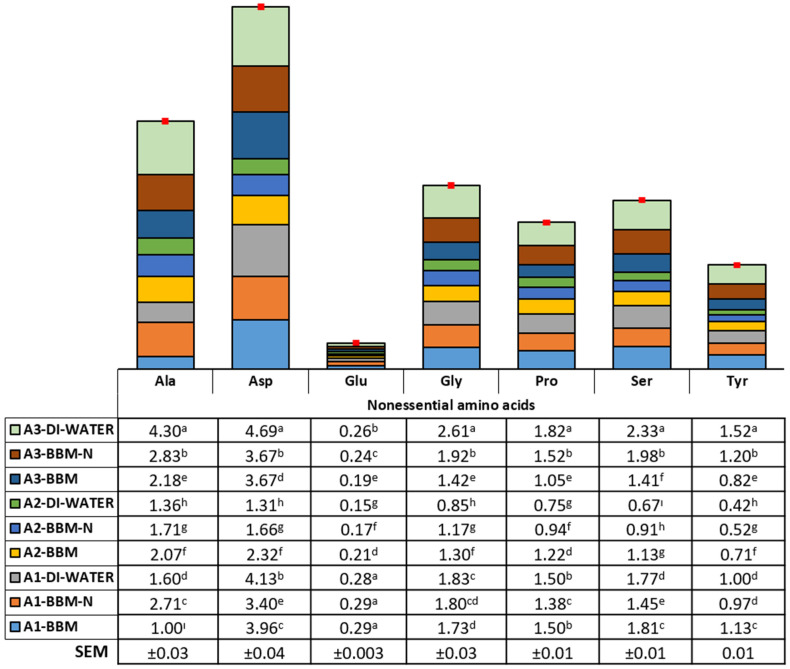
Nonessential amino acid composition of algae species (*Scenedesmus* sp. (A1), *Ankistrodesmus* sp. (A2), and *Synechococcaceae* (A3)) cultivated with poultry house exhaust air. ^a–i^ Differences in letters within columns indicate significant differences among the experimental groups.

**Figure 5 animals-14-00754-f005:**
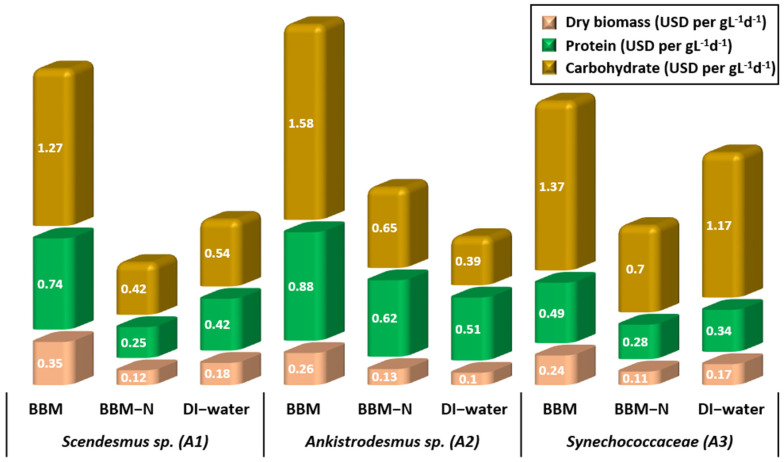
Algal biomass, protein, and carbohydrate production costs of *Scenedesmus* sp., *Ankistrodesmus* sp., and *Synechococcaceae* cultivated with poultry house exhaust air.

**Table 1 animals-14-00754-t001:** ICP-MS operating conditions.

ICP-MS Parameter	Operating Conditions
RF ^1^ power	1550 W
RF matching	1.8 V
Sample depth	8 mm
Lamp–H	0.6 mm
Lamp–V	0 mm
Carrier gas	0.99 mL/min
Nebulizer pump	0.1 rps
S/C temperature	2 °C

^1^ Radio frequency.

**Table 2 animals-14-00754-t002:** Biomass production and biochemical composition of *Scenedesmus* sp., *Ankistrodesmus* sp., and *Synechococcaceae* grown with poultry house exhaust air under different culture conditions.

Exp.	pH *	Temp.* (°C)	CO_2_ * (mgL^−1^d^−1^)	NH_3_ * (mgL^−1^d^−1^)	Initial DW (g L^−1^)	End DW (g L^−1^)	Biomass Productivity (mgL^−1^d^−1^)	Carbohydrate Productivity (mgL^−1^d^−1^)	Protein Productivity (mgL^−1^d^−1^)
A1−BBM	7.92 ± 0.15 ^ef^	14.6 ± 0.06 ^c^	2525.8 ± 11.5 ^b^	3.52 ± 0.02 ^a^	0.22 ^ab^	0.78 ^cd^	23.5 ^bc^	6.5 ^cd^	11.1 ^bc^
A1−BBM−N	7.70 ± 0.01 ^f^	14.7 ± 0.06 ^c^			0.25 ^a^	1.17 ^ab^	43.0 ^a^	12.5 ^a^	20.6 ^a^
A1−DI water	7.73 ± 0.01 ^f^	14.7 ± 0.02 ^c^			0.17 ^bc^	0.58 ^d^	20.0 ^c^	6.7 ^cd^	8.6 ^c^
A2−BBM	8.69 ± 0.06 ^bc^	29.7 ± 0.02 ^a^	1007.8 ± 9.7 ^c^	3.35 ± 0.03 ^b^	0.09 ^de^	1.00 ^bc^	31.3 ^abc^	5.2 ^de^	9.4 ^c^
A2−BBM−N	8.90 ± 0.1 ^b^	29.8 ± 0.04 ^a^			0.12 ^cde^	1.33 ^a^	41.8 ^a^	8.1 ^bc^	8.4 ^c^
A2−DI water	9.20 ± 0.03 ^a^	29.7 ± 0.02 ^a^			0.08 ^e^	1.12 ^ab^	35.9 ^ab^	9.4 ^b^	7.2 ^c^
A3−BBM	8.40 ± 0.05 ^de^	19.7 ± 0.20 ^b^	2774.2 ± 12.6 ^a^	3.24 ± 0.02 ^b^	0.23 ^ab^	0.25 ^e^	33.9 ^abc^	6.0 ^cd^	16.9 ^ab^
A3−BBM−N	8.64 ± 0.46 ^cd^	20.7 ± 0.05 ^b^			0.17 ^bcd^	0.70 ^cd^	46.7 ^a^	7.5 ^bcd^	18.8 ^a^
A3−DI water	8.26 ± 0.09 ^de^	20.9 ± 0.02 ^b^			0.20 ^ab^	0.52 ^de^	21.4 ^bc^	3.1 ^e^	10.8 ^bc^
SEM	±0.09	±0.22	±66.8	±0.05	±0.02	±0.01	±5.35	±0.83	±2.24
*p* VALUES	0.001	0.001	0.0001	0.0001	0.0008	0.0001	0.0015	0.0008	0.0053

* Mean of daily measurements throughout the experiments. ^a–f^ Differences in letters within columns indicate significant differences among the experimental groups. A1 experiments were conducted in January 2023, A2 in June 2023, and A3 in April 2023, A1: *Scenedesmus* sp., A2: *Ankistrodesmus* sp., A3: *Synechococcaceae*; DW: Dry biomass concentration.

**Table 3 animals-14-00754-t003:** Nutrient composition of algae species cultivated with poultry house exhaust air under different culture conditions.

Main Effects	Fatty Acids (%)			Protein (%)	Carbohydrate (%)
	SAFA	MUFA	PUFA		
**Algae species**					
*Scenedesmus* sp.	32.3 ^c^	29.9 ^a^	34.6 ^a^	46.0 ^a^	30.0 ^a^
*Ankistrodesmus* sp.	57.1 ^a^	20.6 ^b^	20.4 ^b^	23.6 ^b^	20.6 ^b^
*Synechococcaceae*	49.9 ^b^	11.4 ^c^	7.9 ^c^	47.0 ^a^	16.4 ^c^
SEM	0.12	0.03	0.03	0.40	0.62
**Culture conditions**					
BBM	45.3 ^b^	21.5 ^b^	19.6 ^c^	17.5 ^b^	21.1 ^b^
BBM−N	44.2 ^c^	21.8 ^a^	22.5 ^a^	21.7 ^a^	21.4 ^b^
DI water	49.8 ^a^	18.6 ^c^	20.8 ^b^	21.7 ^a^	24.6 ^a^
SEM	0.12	0.03	0.03	0.40	0.60
**Algae** **× Culture conditions**					
*Scenedesmus* sp. × BBM	36.0 ± 0.6 ^g^	27.0 ± 0.1 ^c^	32.8 ± 0.1 ^c^	47.3 ± 1.3 ^b^	27.5 ± 0.6 ^b^
*Ankistrodesmus* sp. × BBM	53.5 ± 0.1 ^d^	24.0 ± 0.1 ^d^	20.8 ± 0.1 ^d^	30.3 ± 1.8 ^e^	16.5 ± 0.1 ^cde^
*Synechococcaceae* × BBM	46.5 ± 0.8 ^e^	13.5 ± 0.3 ^h^	5.2 ± 0.1 ^i^	50.4 ± 1.4 ^a^	19.3 ± 0.5 ^c^
*Scenedesmus* sp. × BBM−N	30.8 ± 0.1 ^h^	29.8 ± 0.1 ^b^	36.6 ± 0.1 ^a^	47.7 ± 2.3 ^b^	29.1 ± 0.9 ^b^
*Ankistrodesmus* sp. × BBM−N	57.7 ± 0.5 ^c^	20.2 ± 0.1 ^e^	19.8 ± 0.1 ^f^	20.2 ± 0.1 ^f^	19.2 ± 0.3 ^cd^
*Synechococcaceae* × BBM−N	44.1 ± 0.2 ^f^	15.4 ± 0.1 ^g^	11.0 ± 0.1 ^g^	40.3 ± 1.2 ^d^	16.0 ± 0.7 ^de^
*Scenedesmus* sp. × DI water	30.2 ± 0.1 ^h^	32.8 ± 0.1 ^a^	34.3 ± 0.1 ^b^	42.9 ± 0.8 ^c^	33.4 ± 0.9 ^a^
*Ankistrodesmus* sp. × DI water	60.1 ± 0.2 ^a^	17.5 ± 0.2 ^f^	20.6 ± 0.1 ^e^	20.2 ± 0.2 ^f^	26.2 ± 0.3 ^b^
*Synechococcaceae* × DI water	59.0 ± 0.1 ^b^	5.5 ± 0.01 ^i^	7.5 ± 0.04 ^h^	50.4 ± 0.2 ^a^	14.0 ± 0.9 ^e^
SEM	0.20	0.05	0.06	0.70	1.10
***p*-VALUES**					
Algae species	<0.0001	<0.0001	<0.0001	<0.0001	<0.0001
Culture conditions	<0.0001	<0.0001	<0.0001	<0.0001	<0.0021
Algae × Culture conditions	<0.0001	<0.0001	<0.0001	<0.0001	<0.0001

^a–i^ Differences in letters within columns indicate significant differences among the experimental groups. The lipids, protein, and carbohydrates were expressed as % (mg/100 mg of dry weight of algae biomass).

**Table 4 animals-14-00754-t004:** The heavy metal content of algae species cultivated with poultry house exhaust air under different culture conditions.

Main Effects	Arsenic (mg/kg)	Cadmium (mg/kg)	Mercury (mg/kg)
**Algae species**			
*Scenedesmus* sp.	1.824 ^b^	0.0147 ^c^	0.0136 ^c^
*Ankistrodesmus* sp.	3.047 ^a^	0.0973 ^a^	0.0435 ^b^
*Synechococcaceae*	1.530 ^c^	0.0274 ^b^	0.0564 ^a^
SEM	0.004	0.002	0.006
**Culture conditions**			
BBM	2.420 ^b^	0.0381 ^b^	0.0172 ^b^
BBM−N	2.629 ^a^	0.0588 ^a^	0.0816 ^a^
DI water	1.352 ^c^	0.0425 ^b^	0.0146 ^c^
SEM	0.004	0.002	0.006
**Algae species × Culture conditions**			
*Scenedesmus* sp. × BBM	2.510 ^c^	0.0127 ^h^	0.0124 ^f^
*Ankistrodesmus* sp. × BBM	3.290 ^b^	0.0870 ^b^	0.0206 ^c^
*Synechococcaceae* × BBM	1.460 ^e^	0.0145 ^de^	0.0187 ^cd^
*Scenedesmus* sp. × BBM−N	1.887 ^d^	0.0140 ^e^	0.0176 ^de^
*Ankistrodesmus* sp. × BBM−N	3.500 ^a^	0.1210 ^a^	0.0946 ^b^
*Synechococcaceae* × BBM−N	2.500 ^c^	0.0414 ^c^	0.1330 ^a^
*Scenedesmus* sp. × DI water	1.076 ^f^	0.0174 ^de^	0.0107 ^f^
*Ankistrodesmus* sp. × DI water	2.350 ^c^	0.0839 ^b^	0.0153 ^e^
*Synechococcaceae* × DI water	6.300 ^g^	0.0263 ^d^	0.0179 ^cde^
SEM	0.007	0.004	0.009
***p*-VALUES**			
Algae	0.0001	0.0001	0.0001
Culture conditions	0.0001	0.0001	0.0001
Algae × Culture conditions	0.0001	0.0012	0.0001
**EU permissible limits of heavy metals (mg/kg) in animal feed**	**40 ***	**1 ***	**0.1 ***

^a–h^ Differences in letters within columns indicate significant differences among the experimental groups. * European Union Commission Regulation with regards to maximum levels for certain undesirable substances in animal feed [70].

**Table 5 animals-14-00754-t005:** Color characteristics of algae biomass grown with poultry house exhaust air.

Main Effects	*L** Value	*a** Value	*b** Value	*C**_ab_ Value	*α*° Value
**Algae species**					
*Scenedesmus* sp.	33.6 ^c^	−5.8 ^a^	11.7 ^b^	13.1 ^c^	−1.1 ^b^
*Ankistrodesmus* sp.	38.8 ^a^	−8.5 ^b^	20.9 ^a^	22.6 ^a^	−1.2 ^c^
*Synechococcaceae*	35.7 ^b^	−11.9 ^c^	6.9 ^c^	14.0 ^b^	−0.5 ^a^
SEM	0.11	0.03	0.05	0.05	0.002
**Culture conditions**					
BBM	32.6 ^c^	−8.9 ^b^	10.7 ^c^	14.4 ^b^	−0.85 ^a^
BBM−N	38.0 ^a^	−8.2 ^a^	15.3 ^a^	17.7 ^a^	−1.1 ^c^
DI water	37.5 ^b^	−9.1 ^c^	13.5 ^b^	17.6 ^a^	−0.9 ^b^
SEM	0.11	0.03	0.05	0.05	0.002
**Algae species x Culture conditions**					
*Scenedesmus* sp. × BBM	31.9 ^f^	−5.7 ^b^	8.4 ^g^	10.1 ^h^	−1.0 ^d^
*Ankistrodesmus* sp. × BBM	34.9 ^e^	−9.2 ^f^	17.2 ^c^	19.6 ^c^	−1.1 ^e^
*Synechococcaceae* × BBM	30.8 ^g^	−11.7 ^h^	6.4 ^h^	13.4 ^g^	−0.5 ^b^
*Scenedesmus* sp. × BBM−N	34.5 ^e^	−5.4 ^a^	12.4 ^e^	13.6 ^g^	−1.2 ^f^
*Ankistrodesmus* sp. × BBM−N	39.2 ^c^	−8.4 ^e^	23.7 ^a^	25.1 ^a^	−1.3 ^g^
*Synechococcaceae* × BBM−N	40.2 ^b^	−10.7 ^g^	9.9 ^f^	14.6 ^e^	−0.7 ^c^
*Scenedesmus* sp. × DI water	34.4 ^e^	−6.2 ^c^	14.4 ^d^	15.7 ^d^	−1.2 ^f^
*Ankistrodesmus* sp. × DI water	42.2 ^a^	−7.7 ^d^	21.9 ^b^	23.2 ^b^	−1.3 ^g^
*Synechococcaceae* × DI water	36.0 ^d^	−13.3 ^i^	4.3 ^i^	14.0 ^f^	−0.3 ^a^
SEM	0.19	0.05	0.08	0.09	0.003
***p*-VALUES**					
Algae species	<0.0001	<0.0001	<0.0001	<0.0001	<0.0001
Culture conditions	<0.0001	<0.0001	<0.0001	<0.0001	<0.0001
Algae × Culture condition	<0.0003	<0.0001	<0.0001	<0.0001	<0.0001

^a–i^ Differences in letters within columns indicate significant differences among the experimental group.

**Table 6 animals-14-00754-t006:** Economic estimates of the PBR system run in the poultry house.

Economic Estimates of the PBR System Run in the Poultry House		
**BBM Culture Condition Operating Cost (USD L^−1^ day^−1^)** Nutrients*_$0.046 Mixing and air pumping_$0.034 Illumination energy_$0.0024	**$0.0824 L^−1^ day^−1^**	
**BBM−N Culture Condition Operating Cost (USD L^−1^ day^−1^)** Nutrients*_$0.016 Mixing and air pumping _$0.034 Illumination energy_$0.0024	**$0.0524 L^−1^ day^−1^**	
**DI−Water Culture Condition Operating Cost (USD L^−1^ day^−1^)** Nutrients*_$0 Mixing and air pumping_$0.034 Illumination energy_$0.0024	**$0.0364 L^−1^ day^−1^**	
**PBR System Capital Cost** Air flowmeter_$45.7 Vacuum compressor (1.5 Hp 50 Lt–1.1 kW) _$96 Illumination apparatus_$14.68 LED light bulbs_(14.4 W/m) _$3.5/meter Acrylic sheets (0.47 m^2^/each PBR) _$52.8/PBR Bubble diffuser_$5.56	**$218.24**	

* See Appendix A for the cost calculation of nutrients in Appendix A.

## Data Availability

All data sets collected and analyzed in the study are available on request from the corresponding author. The data are not publicly available due to privacy or ethical restrictions.

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
