# Peer review of "Pollutant Gases to Algal Animal Feed: Impacts of Poultry House Exhaust Air on Amino Acid Profile of Algae"

_animals, 2024, doi:10.3390/ani14050754_

Round 1

Reviewer 1 Report

Comments and Suggestions for Authors

Dear authors,

While the work is interesting there are a few problems to take into account. More data needs to be shown (e.g., a growth curve, DW measurements at the start/end and see for more below). You have this data, but you should also show it. 

Furthermore, the experiment ran for 30 days, but you started with high initial concentrations (?) at least so it seems since you only diluted the start culture 1/2. Did you not have to harvest large parts of the culture during the growth?

There are also issues with the differences between experiments: the temperature, CO2 concentration and NH3 concentration => did this have an effect on the growth? You cannot really state much about this. Now you have results, but it is not really clear what the impact is of these differences in CO2, NH3 and temperature. YOu need to discuss this, highlight that this is a problem and potentially something that influenced your results. You need to mention this. Also: why was there such a big difference in temp, CO2 and NH3?

The work misses some depth to really be more interesting. You should address a few things more (can be done with a few extra sentences, see below also, but e.g., costs price microalgae; current status of algae in animal feed; what the major costs are for producing algae; .....)

a few general remarks:

1° yes, you can use the wastestreams to grow the microalgae, but at the same time (as you also mention in your paper), this would potentially be a problem as your algae might be contaminated with e.g., heavy metals. So perhaps this needs to be highlighted more in the text/in the discussion? A trade off between using algae to grow on 'dirty' streams vs quality of the algae for food/feed. Did you perform an experiment (analysis) on the gas exhaust? Do you know what is in the gas? 

2° At the moment there is data lacking from the paper. You need to provide more data: start DW of the cultures, end DW of the cultures etc. You can add this in an extra table. 

3° table 5 is very hard to follow. Why are there no SD given? Is every measurement only from 1 sample? How the interpret the table?.

4° There is a huge problem with the results as such, you can't really conclude much as the e.g., there are major differences in temperature/CO2 and NH3 concentrations between experiments. How do we know these differences did not have an impact on the results? You do not discuss this. 

5° It is odd that the BBM-N results are so much beter for the growth compared to the BBM condition. What is the difference? THe only difference seems to be that the BBM-N contained no nitrates. So how come the BBM was so much slower in terms of productivity? Was the nitrate a negative factor? 

6° please combine table 2 with table 4. It is too difficult to always switch between the tables. Add table 2 to table 4 to make it more easy to understand/follow the results. + add the species to this table as (e.g., table 1 has to be partly into this new table as well).  1 table with a complete overview is a lot better to understand everything.

7° please make a graph with the growth curves of the algae to better compare the growth.

8° the biggest cost factor for your nutrients is NaNO3, this seems odd that it is 0.03$ for NaNO3 alone. The total cost was 0.046$

9° in general: discuss the price/costs a bit more. The cultivation of microalgae is really expensive and for animal feed almost impossible. YOu need to discuss this a bit more. Also talk a bit more about the prices at large scale etc... One of the major issues with microalgae cultivation are the high prices, but your prices are extremely low....

Please use the attached pdf file for more questions/comments.

Comments on the Quality of English Language

There are a few minor mistakes, see pdf, a few odd sentences (start of the sentence is odd a few times) and there is somewhere written: microalgae's while it can just be microalgae. In general the text is ok.

Author Response

Reviewer#1 comments:

While the work is interesting there are a few problems to take into account. More data needs to be shown (e.g., a growth curve, DW measurements at the start/end and see for more below). You have this data, but you should also show it. Furthermore, the experiment ran for 30 days, but you started with high initial concentrations (?) at least so it seems since you only diluted the start culture 1/2. Did you not have to harvest large parts of the culture during the growth? There are also issues with the differences between experiments: the temperature, CO2 concentration and NH3 concentration => did this have an effect on the growth? You cannot really state much about this. Now you have results, but it is not really clear what the impact is of these differences in CO2, NH3 and temperature. You need to discuss this, highlight that this is a problem and potentially something that influenced your results. You need to mention this. Also: why was there such a big difference in temp, CO2 and NH3?

Response: The investigation of microalgae growth with poultry house exhaust air was carried out for the first time in this study. For this reason, there was a possibility that the poultry house’s exhaust air may have a toxic effect on algal growth or negatively affect algae growth. That’s the reason, the experiments were started with high algal concentrations. No harvesting and no nutrients were added during the experiment to examine the algal growth and how long they would grow without nutrient supplementation or harvesting. In the company where the field study was conducted, a broiler production period lasted about 60 days, including 45 days for rearing and 15 days for cleaning and preparation for the new production period. The poultry house experiments began in January 2024 with experiments involving Scenedesmus sp. Since the subsequent production period started in April, experiments involving the Synechococcaceae species were also conducted. In June, experiments involving Ankistrodesmus sp. were conducted during the following production period. Consequently, the indoor environmental conditions of the poultry house and NH3-CO2 gas concentrations varied in the experiments for each species. Sure, the environmental conditions showed variations during the experimental period, but we aimed to assess the biomass obtained from different microalgae species grown using poultry house exhaust gases. We aimed to compare the specialties of biomass obtained from microalgae species with different BBM, not to compare the environmental factors, as an animal feed. This study focused on creating a new feed ingredient for animal nutrition. So, we have given the environmental conditions as supplementary information.

As you referred, CO2 and NH3 gas concentrations are observed to be at different values during the experiments. The reason for this is that with the increase in the age of the animals in the poultry houses during the feeding period, the gas concentrations in the indoor environment also change. This was expected. For this reason, gas sensors were placed at each PBR tank's inlet and outlet points, and the amount of reduction of gases emitted from the poultry house per unit time was calculated. Another output of our project was to investigate microalgae growth with poultry house exhaust air and calculate their NH3-CO2 gas mitigation efficiencies. For this reason, microalgae growth curves in terms of dry biomass or cell concentrations were not included in this manuscript, and we planned to use them in our next manuscript. However, according to your suggestions, we added the growth curves in this manuscript based on OD (750 nm) values. If you want to see the algal growth graphs for dry biomass or cell concentrations, we can privately share that data with you.

The work misses some depth to really be more interesting. You should address a few things more (can be done with a few extra sentences, see below also, but e.g., costs price microalgae; current status of algae in animal feed; what the major costs are for producing algae; .....)

  1. Comments and/or suggestions: Yes, you can use the wastestreams to grow the microalgae, but at the same time (as you also mention in your paper), this would potentially be a problem as your algae might be contaminated with e.g., heavy metals. So perhaps this needs to be highlighted more in the text/in the discussion? A trade off between using algae to grow on 'dirty' streams vs quality of the algae for food/feed. Did you perform an experiment (analysis) on the gas exhaust? Do you know what is in the gas? 

Response: It is generally known that the gas exhaust of poultry houses comprises dust, pathogens, and harmful gases such as ammonia, nitrous oxide, carbon dioxide, hydrogen sulfide, and methane (Costantino et al., 2020. Reducing gas concentrations in broiler houses through ventilation: Assessment of the thermal and electrical energy consumption, Biosystem Engineering, 199, 135-148.; Mohammad Al-Kerwi et al., 2022. Effects of Harmful Gases Emitted from Poultry Houses on Productive and Health Performance, doi: 10.1088/1755-1315/1060/1/012082). However, the content of the exhaust gas could differ according to the stocking density, environmental conditions, etc. However, in the current study, we focused on the possible usage of microalgae produced with poultry house exhaust gases as a feed ingredient. Therefore, we focused on the content of microbiological load and heavy metal content of algal samples. However, the microbiological analysis is continuing. Therefore, we have only added the heavy metal content. As a result, the last product is algal biomass. Therefore, the analysis of algal samples is important to discuss the possible usage of it as feed. 

  1. Comments and/or suggestions: At the moment there is data lacking from the paper. You need to provide more data: start DW of the cultures, end DW of the cultures etc. You can add this in an extra table.

Response: The initial and last day of dry biomass concentration was provided in Table 2 according to your suggestion.

  1. Comments and/or suggestions: Table 5 is very hard to follow. Why are there no SD given? Is every measurement only from 1 sample? How the interpret the table?

Response: In Table 5, SEM is given in the Table.

  1. Comments and/or suggestions: There is a huge problem with the results as such, you can't really conclude much as the e.g., there are major differences in temperature/CO2 and NH3 concentrations between experiments. How do we know these differences did not have an impact on the results? You do not discuss this. 

Response:  As mentioned below, This study was carried out under field conditions in one poultry house with an ordinary. Sure, the environmental conditions showed variations during experimental period, but we aimed to assess the biomass obtained from different microalgae species grown with using poultry house exhaust gases. We aimed to compare the specialties of biomass obtained from microalgae species with different BBM, not to compare the environmental factors, as an animal feed. This study focused on creating a new feed ingredient for animal nutrition. So, we have given the environmental conditions as supplementary information. Also, we have added some discussion part about the effecting factors of biomass productivity in microalgae.

  1. Comments and/or suggestions: It is odd that the BBM-N results are so much beter for the growth compared to the BBM condition. What is the difference? THe only difference seems to be that the BBM-N contained no nitrates. So how come the BBM was so much slower in terms of productivity? Was the nitrate a negative factor? 

Response: In our previous study (Uguz et al., 2022), we cultivated algae with swine house exhaust air using BBM. However, in that study, we observed a mismatch of NH3 supply versus demand for algal growth. During periods when NH3 was in excess, the gas was stored in the growth medium due to its high water solubility. A substantial bypass of NH3 through the PBR happened only when the medium became saturated. A key to air pollutant removal by PBRs is matching the N demand by algae with the N supply (NH3) from the polluted air. That’s why we eliminated the N from BBM to provide N-source to algae from poultry exhaust air.

Reference:

Uguz, S., Anderson, G., Yang, X., Simsek, E., & Osabutey, A. (2022). Cultivation of Scenedesmus dimorphus with air contaminants from a pig confinement building. Journal of Environmental Management, 314, 115129.

  1. Comments and/or suggestions: please combine table 2 with table 4. It is too difficult to always switch between the tables. Add table 2 to table 4 to make it more easy to understand/follow the results. + add the species to this table as (e.g., table 1 has to be partly into this new table as well).  1 table with a complete overview is a lot better to understand everything.

Response: Table 1, 2 and 4 were combined according to your comment.

  1. Comments and/or suggestions: please make a graph with the growth curves of the algae to better compare the growth.

Response: Algal growth curves of the poultry house experiments were provided in Figure 2 as your suggestion.

  1. Comments and/or suggestions: the biggest cost factor for your nutrients is NaNO3, this seems odd that it is 0.03$ for NaNO3 alone. The total cost was 0.046$

Response: The cost of the nutrients was calculated based on the amount of chemicals used for BBM. Detailed calculation of the BBM is given in the Table below. The total cost of the 1 L BBM is $0.14, and the cost of NaNO3 is $0.09. These are for 1 L BBM. We multiply this amount by 10 L working volume and then divide the experiment duration (30 days) to find the USD cost per liter per day. We can try to explain if you have further questions about these calculations.

Table 1. The cost calculations of 1 L Bold’s Basal Medium

Chemicals

The amount used for 1 L BBM (mg/Ld)

Price of each chemicals

Cost Turkish liras (TL)

Cost dollars

KHPO4

175 mg/l

300 TL/1000 g

0,053

0,00654

CaCl2*2H2O

25 mg/l

115,15 TL/1000 g

0,003

0,00037

MgSO4*7H2O

75 mg/l

726,41 TL/1000 g

0,054

0,00667

NaNO3

250 mg/l

751.22 TL/250 g

0,75

0,09259

K2HPO4

75 mg/l

253,45 TL/100 g

0,19

0,02346

MoO3

1.42 mg/l

902,61 TL /100 g

0,013

0,00160

NaCL

25 mg/l

31,98 TL/1000 g

0,0008

0,00010

EDTA C10H16N2O8

50 mg/l

347 TL/500 g

0,035

0,00432

KOH

31 mg/l

103,62 TL/1000 g

0,0032

0,00040

FeSO4*7H2O

4.98 mg/l

142,87 TL/1000 g

0,0007

0,00009

H2SO4

1uL

130,41 TL/2500 mL

0,00005

0,00001

H3BO3

11.42 mg/l

179,42 TL/100 g

0,02

0,00247

Trace Metal Solution  (1 mL/L)

1 mL

12,95 TL/1000 mL

0,013

0,00160

Total cost for 1 L BBM

0,140216

  1. Comments and/or suggestions:  in general: discuss the price/costs a bit more. The cultivation of microalgae is really expensive and for animal feed almost impossible. YOu need to discuss this a bit more. Also talk a bit more about the prices at large scale etc... One of the major issues with microalgae cultivation are the high prices, but your prices are extremely low....

Response: The economic estimates of the PBR system and scaling up cost effects were discussed in Lines 565-575. Our prices is not low compared to the literature. Our costs reported in Table 6 was the operating costs for per liter per day. The dry biomass production costs based on the biomass productivities in this study (see Table 2) are in the range of $1014 to $2632 per kg dry biomass. The dry biomass production costs were reported as between $44 to $108.6 per kg dry weight by Vazquez-Romero et al. [67] as discussed in Lines 567-575. However, the production costs in this study was for only lab-scale PBR sytem. We just would like to give some cost information to effect of algae species and culture conditions on the production costs.

Thank you very much for your consideration.

Best regards,

Seyit Uguz
Research Assistant,
Biosystems Engineering Depart. | Faculty of Agriculture | Bursa Uludag University
+90 224 294 1617 | [email protected]
Görükle Campus, 16059, Nilufer, Bursa/Turkey

Reviewer 2 Report

Comments and Suggestions for Authors

Attached

Comments on the Quality of English Language

Manuscript Title: Pollutant Gases to Algal Animal Feed: Impacts of Poultry House Exhaust Air to Amino Acid Profile of Algae

My comments 

Abstract 

Line 24-26: The sentence “Poultry operations, while crucial for the global food supply, contribute to environmental concerns, particularly in terms of ammonia and carbon dioxide gas emissions.” should be revised for a proper meaning. 

Line 47: The “Keywords” should be replaced as “Keywords: Pollutant gas; Microalgae; Animal feed; Sustainability; Amino acids; Heavy metals”.

Introduction 

Line 73: Provide a suitable reference for the sentence “In addition to their nutritional value, algae have been recognized for their potential to reduce the environmental impacts of animal production.”

Provide more information concerning “Pollutant Gases to Algal Animal Feed”. 

Line 118-128: Revise the entire part and clearly define the objective of the work.

2. Materials and Methods

Line 147: The sentence “Algae cultures were periodically doubled…” is not clear. 

Line 149: What is the composition of poultry exhaust air?

Line 151: Why PBRs were continuously aerated with poultry exhaust air at a rate of 0.5 L min-1 per liter of PBR volume ? What is the basis of choosing air flow rate of 0.5 L min-1 per liter ?

Line 187: Explain table 2 data in text also.

3. Results and Discussions

Line 277: The subheading “Nutritional composition” should be renamed as “Effect of Nutritional composition”.

Line 278-286: Replace this part in the “introduction part”.

Line 291: Remove word like “argued”.

Line 302: Explain the significance of “carbohydrate productivity” with respect to this study. 

Line 314: Explain the significance of “protein productivity” with respect to this study.

Line 341-344: Replace this part in the “introduction part”.

Line 371-378: Rewrite the entire paragraph for a proper meaning. 

Line 396-400: Explain the figure 2 and 3 in more detail. 

Line 420: What is the source of “heavy metals” in this study?

Line 440: Explain Table 6 in the text.

Line 445: What is the significance of the color characteristics of algae concerning this study? 

Line 463: Explain Table 7 in the text.

Line 465: Explain “economic estimates of PBR system” in more detail. 

Line 478: Explain Table 8 in the text.

Line 505: Explain Figure 3 with more detail in the text

4. Conclusions

Line 508: The heading “5. Conclusions” should be revised as “4. Conclusions”.

 Line 511-513: The sentence “It could be concluded that the feed material produced by microalgae could be efficiently used due to its chemical composition to enhance the nutritional composition of meat and eggs through partial replacement of conventional dietary protein sources” should be revised for a proper meaning. 

Line 515-516: The sentence “Nevertheless, it is important to emphasize that the nutrient availability of different microalgae species must be considered to determine their potential usage in animal nutrition” should be revised for a proper meaning. 

Author Response

Reviewer#2 comments:

Abstract 

  1. Comments and/or suggestions: Line 24-26: The sentence “Poultry operations, while crucial for the global food supply, contribute to environmental concerns, particularly in terms of ammonia and carbon dioxide gas emissions.” should be revised for a proper meaning. 

Response:  The sentence has been revised as: “Poultry production is crucial for the global food supply but contributes to environmental concerns, particularly in terms of ammonia and carbon dioxide gas emissions.”

  1. Comments and/or suggestions: Line 47: The “Keywords” should be replaced as “Keywords: Pollutant gas; Microalgae; Animal feed; Sustainability; Amino acids; Heavy metals”.

Response: The correction has been made for keywords.

Introduction 

  1. Comments and/or suggestions: Line 73: Provide a suitable reference for the sentence “In addition to their nutritional value, algae have been recognized for their potential to reduce the environmental impacts of animal production.”

Response:  References have been added into the sentence as: “In addition to their nutritional value, algae have been recognized for their potential to reduce the environmental impacts of animal production (Toledo-Cervantes et al., 2018; Wu et al., 2023).”

  1. Comments and/or suggestions: Provide more information concerning “Pollutant Gases to Algal Animal Feed”..

Response:  The sentence has been revised as: “Furthermore, the economic viability of algae production has been underscored, particularly in utilizing algae biomass for animal feed, highlighting the potential for pollutant gases released from the poultry houses, for example ammonia, nitrous oxide, carbon dioxide, hydrogen sulphide, and methane (Costantino et al., 2020) and waste utilization in feed management [22]”.

  1. Comments and/or suggestions: Line 118-128: Revise the entire part and clearly define the objective of the work.

Response: The objective of the study has been revised as: “This study was focused on the possible usage of the poultry exhaust air and different culture conditions (BBM, BBM-N, DI-water) to culture microalgae, to compare the various parameters including algae productivity, chemical composition, and cost analysis of three different algae species, namely Scenedesmus sp. (AQUAMEB-60), Ankistrodesmus sp. (AQUAMEB-33), and Synechococcaceae (AQUAMEB 32), and subsequently the possibility usage of these microalgaal biomass as a feed ingredient in animal nutrition.”

Materials and Methods

  1. Comments and/or suggestions: Line 147: The sentence “Algae cultures were periodically doubled…” is not clear. 

Response: Algal cultures were doubled weekly with autoclaved BBM to reach the 10 L volume for the experiments. This information in Lines 160-161 was revise according to your comment.

  1. Comments and/or suggestions: Line 149: What is the composition of poultry exhaust air?

Response: It has been indicated in the sentence as “….poultry exhaust air (mainly CO2 and NH3) and…”

  1. Comments and/or suggestions: Line 151: Why PBRs were continuously aerated with poultry exhaust air at a rate of 0.5 L min-1 per liter of PBR volume ? What is the basis of choosing air flow rate of 0.5 L min-1 per liter ?

Response: PBRs were continuously aerated with poultry exhaust air to reduce NH3 and CO2 gases from the barn air by photobioreactor systems. The higher aeration can help to capture more gases from the barn air but on the other hand, higher aeration may damage the algal cells. That is the reason that aeration rate is really important for algal cells. In our previous study, we investigated the S. Dimorphus growth with swine house exhaust air aerated at the rate of 0.5 vvm. Also other lab scale studies conducted by Kang and Wen (2015) and Kang (2012) were used the similar aeration rates. For this reason, we chose the aeration rates used in the literature and based on our experience in our previous studies.

  1. Comments and/or suggestions:  Line 187: Explain table 2 data in text also.

Response: Table was explained in Line 205-219 as your comment.

Results and Discussions

  1. Comments and/or suggestions: Line 277: The subheading “Nutritional composition” should be renamed as “Effect of Nutritional composition”.

Response: The subheading has been revised as: “Comparison of Nutritional composition of microalgal biomass”

  1. Comments and/or suggestions: Line 278-286: Replace this part in the “introduction part”

Response: It has been placed into the introduction part.

  1. Comments and/or suggestions: Line 291: Remove word like “argued”.

Response: “indicated” has been added instead of “argued”.

  1. Comments and/or suggestions: Line 302: Explain the significance of “carbohydrate productivity” with respect to this study.

Response: This has been added into the manuscript.

  1. Comments and/or suggestions: Line 314: Explain the significance of “protein productivity” with respect to this study.

Response: The significance of protein productivity was explained in Lines 365-370.

  1. Comments and/or suggestions: Line 341-344: Replace this part in the “introduction part”.

Response: This part should be stay in the results and discussion part. Because it states a finding obtained from the study.

  1. Comments and/or suggestions: Line 371-378: Rewrite the entire paragraph for a proper meaning.

Response: The paragraph has been revised as: “When regarding to amino acid profile of micro-algal biomass, it could be suggested that microalgae’s could have an opportunity to be an alternative for conventional protein sources in animal nutrition. This is similar with previous reports explained by Burja et al. [54], Jacob-Lopes et al. [55], and Uguz and Sozcu [25]. Regarding to both essential amino acids and non-essential amino acids, Synechococcaceae grown with DI-water could be offered as an amino acid source in animal nutrition. In a previous report, it has been highlighted that different species of microalgae could be used in ab combination to provide a more balanced amino acid profile [48].”

  1. Comments and/or suggestions: Line 396-400: Explain the figure 2 and 3 in more detail. 

Response: This has been done.

  1. Comments and/or suggestions: Line 420: What is the source of “heavy metals” in this study?

Response: Arsenic (As), lead (Pb), cadmium (Cd), and mercury (Hg) have been recognized as most toxic heavy metals that are continuously released into the environment, both from natural sources and from anthropogenic production of fertilizers, industrial activities, and waste disposal (Aljohani, 2023). Therefore, we needed to check the heavy metal content of biomass, because we wanted to see the possibility usage of microalgae as an animal feed ingredient.

  1. Comments and/or suggestions: Line 440: Explain Table 6 in the text.

Response: Table 6 was explained in Lines 475-503.

  1. Comments and/or suggestions: Line 445: What is the significance of the color characteristics of algae concerning this study? 

Response: We want to see the specialties of microalgae biomass as animal feed ingredient. As you know, the color characteristics of feed is an important issue for the color of animal origin food, for example, egg yolk color or meat color. Therefore, we measured the color characteristics.

  1. Comments and/or suggestions: Line 463: Explain Table 7 in the text.

Response: Table 7 was explained in Lines 510-515 and discussed in Lines 516-526.  

  1. Comments and/or suggestions: Line 465: Explain “economic estimates of PBR system” in more detail.

Response: The economic estimates of the PBR system was detailed and also discussed in the Lines 565-575. We can revise it if you have further suggestion or comments specifically.

  1. Comments and/or suggestions: Line 478: Explain Table 8 in the text.

Response: Table 8 was explained in Lines 530-541.  

  1. Comments and/or suggestions: Line 505: Explain Figure 3 with more detail in the text

Response: Figure 5 was explained in Lines 540-546 and 551-555.

Conclusions

  1. Comments and/or suggestions: Line 508: The heading “5. Conclusions” should be revised as “4. Conclusions”.

Response: It is revised as “4. Conclusions”.

  1. Comments and/or suggestions: Line 511-513: The sentence “It could be concluded that the feed material produced by microalgae could be efficiently used due to its chemical composition to enhance the nutritional composition of meat and eggs through partial replacement of conventional dietary protein sources” should be revised for a proper meaning. 

Response: The sentence has been revised as: “The interest in microalgae as an alternative food and/or feed material is expected to increase due to the demand for plant-based products and the healthy nutritional characteristics of microalgae. It could be affirmed that the feed material obtained from microalgae could be efficiently used due to its nutritional composition through partial replacement of conventional dietary protein sources.”

  1. Comments and/or suggestions: Line 515-516: The sentence “Nevertheless, it is important to emphasize that the nutrient availability of different microalgae species must be considered to determine their potential usage in animal nutrition” should be revised for a proper meaning.

Response: The sentence has been revised as: “Nevertheless, the nutrient availability of different microalgae species must be considered to determine their potential usage in animal nutrition”.

Thank you very much for your consideration.

Best regards,

Seyit Uguz
Research Assistant,
Biosystems Engineering Depart. | Faculty of Agriculture | Bursa Uludag University
+90 224 294 1617 | [email protected]
Görükle Campus, 16059, Nilufer, Bursa/Turkey

Reviewer 3 Report

Comments and Suggestions for Authors

Reviewers comments:

Article Pollutant Gases to Algal Animal Feed: Impacts of Poultry House Exhaust Air to Amino Acid Profile of Algae

General comments:

Line 56: what does gamered means? Please correct

Please explain in Materials and Methods how flat panels were harvested or diluted during the 30 days of continuous operation.

Have you noticed any toxic effect of ammonia on algae? Table 2 suggests pH was close to nine and sometimes even higher. Did it affect negatively algal species You used? Please explain.

Line 278 - 279 Sentence is unclear. Please reformulate

Table 6. As, Cd and Hg levels are rather high in biomass, although lower than EU regulative for animal feed.

Nevertheless, there is an obvious issue with algal natural ability to intake heavy metals. The secuestration of these highly toxic compounds from the exhaust gas and its further bioaccumulation in the animals that are feed with algae meals can be a problem.

Please comment on this and add few sentences in the text on how you see this problem.

Please state clearly in the Conclusions section which species is the best and has potential to be cultivated with exhaust gas and under which condition.

Text is to large and it can be optimized, in my opinion.

Author Response

Reviewer#3 comments:

  1. Comments and/or suggestions: Line 56: what does gamered means? Please correct

Response: It was corrected as your suggestion.

  1. Comments and/or suggestions: Please explain in Materials and Methods how flat panels were harvested or diluted during the 30 days of continuous operation.

Response: Harvesting method and dilution was provided in Lines 198-199.

  1. Comments and/or suggestions: Have you noticed any toxic effect of ammonia on algae? Table 2 suggests pH was close to nine and sometimes even higher. Did it affect negatively algal species You used? Please explain.

Response:  We haven’t seen any toxic effect of ammonia on algae. In our previous study (Uguz et al., 2022), we cultivated algae with swine house exhaust air and also did not observe any contamination for algal growth. In this study, no pH adjustment was made in the culture medium. And It can be seen that pH of E4-E9 experiments was reached to 8 and 9. This is possibly because of that the medium becomes basic when NH3 absorbed into the growth medium exceeds that taken up by algae. The pH of the cultivation medium was projected to be a concern because of high NH3 concentrations. However, the measured pH values mostly stayed in the optimal range for algal growth, and the pH fluctuations caused by the pit air were minor and could be easily regulated.

Reference:

Uguz, S., Anderson, G., Yang, X., Simsek, E., & Osabutey, A. (2022). Cultivation of Scenedesmus dimorphus with air contaminants from a pig confinement building. Journal of Environmental Management314, 115129.

  1. Comments and/or suggestions: Line 278 - 279 Sentence is unclear. Please reformulate

Response: The sentence in Lines 294-295 was revised according to your comment.

  1. Comments and/or suggestions: Table 6. As, Cd and Hg levels are rather high in biomass, although lower than EU regulative for animal feed.

Response: Analysis of micro-algal biomass in this study indicated that the heavy metal content are well within the prescribed safety limits according to EU regulative for animal feed.

  1. Comments and/or suggestions: Nevertheless, there is an obvious issue with algal natural ability to intake heavy metals. The secuestration of these highly toxic compounds from the exhaust gas and its further bioaccumulation in the animals that are feed with algae meals can be a problem. Please comment on this and add few sentences in the text on how you see this problem.

Response: This part has been added: “It is known that microalgae have capability to absorb potentially harmful heavy metals with different pathways including bio-sorption, biotransformation or bioaccu-mulation (Leong and Chang, 2020 ). Therefore, the heavy metal content of microalgae shows variation according to the cultivation conditions (Nagarajan et al., 2021 ). Due to accumulating of heavy metals by microalgae, the usage of its in food or feed chain could cause negative effects on human or animal health. In this respect, before application of microalgae biomass as a nutritional material, the analysis of heavy metal, mainly Pb, Cd, Ar and Hg, have to be provided and guaranteed for their safety (Radin et al., 2017) . However, Kay and Barton (199 1) hypothesized heavy metals originated from micro algae could not effectively absorbed by the gastro-intestinal tract of animal and any strict proof for this issue.”

  1. Comments and/or suggestions: Please state clearly in the Conclusions section which species is the best and has potential to be cultivated with exhaust gas and under which condition. Text is to large and it can be optimized, in my opinion.

Response: It is an important issue for safety usage in animal production. Therefore, Scenedesmus sp. has many more advantages for its’ nutritional content and safety usage and has the potential to be cultivated with exhaust gas.

Thank you very much for your consideration.

Best regards,

Seyit Uguz
Research Assistant,
Biosystems Engineering Depart. | Faculty of Agriculture | Bursa Uludag University
+90 224 294 1617 | [email protected]
Görükle Campus, 16059, Nilufer, Bursa/Turkey

Round 2

Reviewer 1 Report

Comments and Suggestions for Authors

Dear authors,

The manuscript has improved substantially, but please address the following minor issues:

- Add a reference for: The quest for sus-53 tainable poultry production has led to exploring alternative feed materials without any 54 chemical residue and non-genetically modified. The utilization of algae in poultry feed 55 has attracted significant attention due to its potential to enhance the sustainability and 56 nutritional quality of poultry production. 

- Add an extra, more correct reference for (the current reference is not good enough): The 69 nutritive value of specific algae, such as Spirulina and Laminaria digitata, has been studied 70 in the context of poultry feed, with findings indicating their potential to influence broiler 71 production performance, meat quality, and lipid profile [8]. 

- Try to add a reference more specific on chicken here (the current one is more about ducks, which is also poultry, but there are more chicken than ducks being bred for food): However, the environmental impact of poultry produc-76 tion, particularly the emission of ammonia and carbon dioxide gases, has become a grow-77 ing concern [12].

- Add an extra, better reference here (the current one is too specific about biodiesel), you can add a general review that also talks about the cultivation costs of microalgae: Today, commercial production of microalgae could not be cost-competitive, because 123 it demands a higher amount of water, energy and nutrients [21]. 

- Line 167: you state that the algae were grown in triplicate, but why do you not provide a SD or SEM value per measurement? E.g. in table 2 you mention the pH or Temp that you measured, but you do not show the SD or SEM per measurement. Normally if you replicate (triplicate here) your experiments you have have a SD or SEM. You measured 3 pH values each time and not 1. 

There is only a SEM at the bottom of the table , but this is the SEM over all the measurements (over all the experiments combined) which is ok, but you also have to add the SD or SEM per measurement.

Table 2 should state: A1-BBM ph= 7.92 +- X (you have to add this SD or SEM per value, not just at the bottom of the column)

And this for every measurement. 

=> in your first original table you did this (it had for example: 

7.92±0.15 

and now the table just states 7.92)

Did you remove the SD/SEM values to make sure all the data fits in the table (on the page)? If this is the case: provide a more complete table with the SD /SEM values in the supplement or turn the table around (that it goes from the top of the page to the bottom, rather than from left to right; put the page in 'landscape' view)

Also the CO2 and the NH3: why is there no SD or SEM? We do not know now what the differences were in measurements and how often was CO2 or NH3 measured?

- Table 2: 0,58 is shown in the table, adjust to 0.58

- Table 2: what is E1-E3 ? Check this below the table. There is no E1-E3 in the table, correct this.

- What does this mean (factor 2-25?): Van Krimpen et al. [44] indicated that the protein yield of microalgae is higher 365 than conventional protein sources in North Western Europe according to the 2-25 factors. 

- Table 1 in your reply letter where you show the cost calculation: perhaps add this table to your paper as a supplemental table? It makes it easier to see how you calculated the costs. You can add for example see also Suppl T1 to table 6 in your paper

- Line 455: why do you write microalgae's? Is it not just microalgae? Also start the sentence simply with 'regarding'.. The 'when' is not needed.

- Table 3: the data shown there, e.g. the SAFA values: is this 1 measurement? Or ? I assume the the values at the top (for the 3 species) is the average of this species over all the tests? 

Did you measure these SAFA, MUFA contents etc not in triplo? If so where is the SD/SEM value for each measurement? 

- please check lines 571 to 575. You state that the production costs INCREASED from 294.84 to 2948 metric ton of harvested biomass, but then you state that the costs decreased when going from 10 to 100 ha? I am not sure I understand it, first you stated that the costs increased and then you state they decreased?

Comments on the Quality of English Language

A few errors throughout the text, very minor.
